# Learning Multi-Agent Coordination for Enhancing Target Coverage in Directional Sensor Networks

**Jing Xu**[* 1, 6], **Fangwei Zhong**[* 2, 3, 5], **Yizhou Wang**[2, 4]

[1] Center for Data Science, Peking University
[2] Dept. of Computer Science, Peking University
[3] Adv. Inst. of Info. Tech, Peking University
[4] Center on Frontiers of Computing Studies, Peking University
[5] Advanced Innovation Center For Future Visual Entertainment, Beijing Film Academy
[6] Deepwise AI Lab
jing.xu@pku.edu.cn, zfw@pku.edu.cn, yizhou.wang@pku.edu.cn

## Abstract

Maximum target coverage by adjusting the orientation of distributed sensors is an important problem in directional sensor networks (DSNs). This problem is challenging as the targets usually move randomly but the coverage range of sensors is limited in angle and distance. Thus, it is required to coordinate sensors to get ideal target coverage with low power consumption, e.g. no missing targets or reducing redundant coverage. To realize this, we propose a Hierarchical Target-oriented Multi-Agent Coordination (HiT-MAC), which decomposes the target coverage problem into two-level tasks: targets assignment by a *coordinator* and tracking assigned targets by *executors*. Specifically, the coordinator periodically monitors the environment globally and allocates targets to each executor. In turn, the executor only needs to track its assigned targets. To effectively learn the HiT-MAC by reinforcement learning, we further introduce a bunch of practical methods, including a self-attention module, marginal contribution approximation for the coordinator, goal-conditional observation filter for the executor, etc. Empirical results demonstrate the advantage of HiT-MAC in coverage rate, learning efficiency, and scalability, comparing to baselines. We also conduct an ablative analysis on the effectiveness of the introduced components in the framework.

## 1 Introduction

We study the target coverage problem in Directional Sensor Networks (DSNs). In DSNs, every node is equipped with a "directional" sensor, which perceives a physical phenomenon in a specific orientation. Cameras, radars, and infrared sensors are typical examples of directional sensors. In some real-world applications, the sensors in DSNs are required to dynamically adjust their own orientation to track mobile targets, such as automatically capturing sports game videos[1], actively tracking interesting objects [1]. To realize these applications, the target coverage acts as a crucial point, which puts emphasis on **how to cover the maximum number of targets with the finite number of directional sensors**. It is challenging as the targets usually move randomly but the locations of sensors are fixed. Meanwhile, the coverage range for sensors is limited in angle and distance. To do this, it is required to collaboratively adjust the orientation of each sensor in DSNs by a multi-agent system to cover targets. In practice, the multi-agent system for DSNs should : 1) accomplish the global task via

multi-agent collaboration/coordination 2) be of good generalization to different environments 3) be low-cost in communication and power consumption.

In this paper, we are interested in building such a multi-agent collaborative system via multi-agent reinforcement learning (MARL), where the agents are learned by trial and error. The simplest way is to build a centralized controller to globally observe and control the DSNs simultaneously. And we can formulate it as a single-agent RL problem and directly optimize the controller by the off-the-shelf algorithms [2, 3]. However, it is usually infeasible in real-world scenarios. It is because that the system highly relies on real-time communication between the controller and sensors. Moreover, it is hard to further extend the system for large scale networks, as the computational cost in the server will be dramatically expanded with the increasing agent numbers. Recently, the RL community has taken great efforts on learning a fully decentralized multi-agent collaboration [4–6] for various applications, e.g. playing real-time strategy games [7], controlling traffic light [8], self-organizing swarm system[9]. In a decentralized system, each agent runs individually, which observe the environment by themselves and exchange their information by peer-to-peer communication. Such a decentralized system could run on a large scale multi-agent system and be low-cost on communication (even without communication). But in most cases, the distributed policy is unstable and difficult to learn, as they usually affect others leading to a non-stationary environment. Even though this issue has been mitigated by the recent centralized training and decentralized execution methods [4–6], a remaining open challenge is how to effectively train a centralized critic to decompose the global reward to each agent for learning the optimal distributed policy, i.e. multi-agent credit assignment problem [10, 11]. To this end, we are motivated to explore a feasible solution to combine the advantages of above methods to learn a multi-agent system for the target coverage problem effectively.

We propose a Hierarchical Target-oriented Multi-agent Coordination framework (HiT-MAC) for the target coverage problem, inspired by the recent success in Hierarchical Reinforcement Learning (HRL) [12–14]. This framework is a two-level hierarchy, composed of a centralized *coordinator* (high-level policy) and a number of distributed *executors* (low-level policy), shown as Fig. 1. While running, (a) the *coordinator* collects the observations from executors and allocates goals (a set of targets to track) for each executor, and (b) each *executor* individually takes primitive actions to complete the given goal for $k$ time steps, i.e. tracking the assigned targets. After the $k$ steps execution, the coordinator is activated again. Then, steps a and b iterate. In this way, the target coverage problem in DSNs is decomposed into two sub-tasks at different temporal scales. Both coordinator and executors can be trained by the modern single-agent reinforcement learning method (e.g. A3C [2]) to maximize expected future team reward (coordinator) and goal-conditioned rewards (executors), respectively. Specifically, the team reward is factored by the coverage rate; the goal-conditioned reward is about the performance of a sensor to track the selected targets, measured by the relative angle among sensor and target. So, it can also be considered as the cooperation between the coordinator and executors.

To implement a scalable HiT-MAC, there are two challenges to overcome: (1) For the coordinator, how to learn a policy to handle the assignment among variable numbers of sensors and targets? (2)

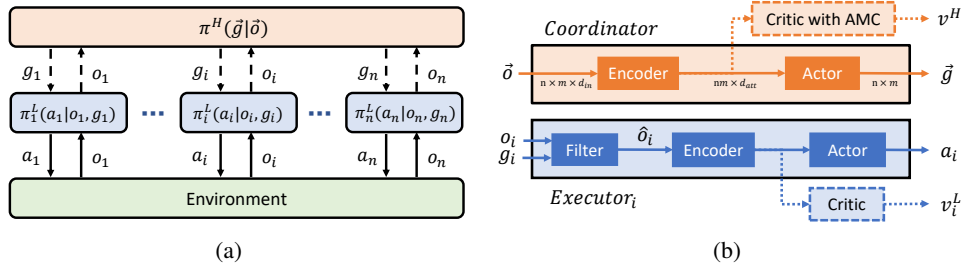

(a)          (b)

Figure 1: An overview of the HiT-MAC framework. Fig. 1(a) is the two-level hierarchy of HiT-MAC. Periodically (every $k$ steps), the high-level policy (*coordinator*) $\pi^H(\vec{g}|\vec{o})$ collects joint observation $\vec{o} = (o_1, \ldots, o_n)$ from sensors and distributes target-oriented goal $g_i$ to each low-level policy (*executor*). In turn, the executor $\pi_i^L(a_i|o_i, g_i)$ directly interacts with the environment to track its own targets. The observation $o_i$ describes the spatial relation between sensor $i$ and targets. The goal $g_i$ allocates the targets to be followed by the executor $i$. Note that the solid line and the dashed line are executed at every step and every $k$ steps respectively. Fig. 1(b) is the details of the coordinator and executor. The critics for them are only used while training the networks. Please refer to Sec. 3 for more details.

For the executor, how to train a robust policy that could perform well in any possible cases, e.g. given different target combinations? Hence, we employ a battery of practical methods to address these challenges. Specifically, we adopt the self-attention module to handle variable input size and generate a order-invariant representation. We estimate values by approximating the marginal contribution (AMC) of each pair of the sensor-target assignments. With this, the critic could better estimate and decompose the team value in a more accurate way, which guides to a more effective coordination policy. For the executor, we further introduce a goal-conditioned filter to remove the observation of the irrelative targets and a goal generation strategy for training.

We demonstrated the effectiveness of our approach in a simulator, comparing with the state-of-the-art MARL methods, and a heuristic method. To be specific, our method achieves the highest coverage ratio and fastest convergence in the case of $4$ sensors and $5$ targets. We also validate the good transferability of HiT-MAC in environments with different numbers of sensors ($2 \sim 6$) and targets($3 \sim 7$). Besides, we also conduct an ablation study to analyze the contribution of each key component.

Our contributions can be summarized in three-folds:

- We study the target coverage problem in DSNs and propose a Hierarchical Target-oriented Multi-agent Coordination framework (HiT-MAC) for it. To the best of our knowledge, it is the first hierarchical reinforcement learning method for this problem.

- A bunch of practical methods is introduced to effectively learn a generalizable HiT-MAC, including a self-attention module, marginal contribution approximation, goal-conditioned filter, and so on.

- We release a numerical simulator to mimic the real scenario and conduct experiments in the environments to illustrate the effectiveness of the introduced framework.

## 2  Preliminary

**Problem Definition.** The target coverage problem considers how to use a number of active sensors to continuously cover maximum number of targets. In this case, there are $n$ sensors and $m$ mobile targets in the environment. Sensors are randomly placed in the environment with limited coverage range. The targets randomly walk around the environment. A target is covered by the sensor network, once it is monitored by at least one sensor. The orientation of the sensor is adjustable, but the changing angle at each step is restricted considering the physical constraints. Besides, considering the efficiency problem, every movement will take additional cost in power.

**Dec-POMDPs.** It is natural to formulate the target coverage problem in $n$ sensors networks as a Dec-POMDPs [15]). It is governed by the tuple $\langle N, S, \{A_i\}_{i \in N}, \{O_i\}_{i \in N}, R, Pr, Z \rangle$ where: $N$ is a set of $n$ agents, indexed by $\{1, 2, ..., n\}$; $S$ is a set of world states; $A_i$ is a set of primitive actions available for agent $i$, and forming joint actions $\vec{a}_t = (a_{1,t}, ..., a_{n,t})$ with others; $O_i$ is the observation space for agent $i$, and its local observation $o_{i,t} \in O_i$ is drawn from observation function $Z(o_{i,t}|s_t, \vec{a}_t)$; $R : S \rightarrow R$ is the team reward function, shared among agents; $Pr : S \times A_1 \times ... A_n \times S \rightarrow [0, 1]$ defines the transition probabilities between states over joint actions. Notably, the subscript $t \in \{1, 2, ...\}$ denotes the time step. At each step, each agent acquires observation $o_{i,t}$ and takes an action $a_{i,t}$ based on its policy $\pi_i(a_{i,t}|o_{i,t})$. Influenced by the joint action $\vec{a}_t$, the state $s_t$ is updated to a new state $s_{t+1}$ according to $Pr(s_{t+1}|s_t, \vec{a}_t)$. Meanwhile, the agent $i$ receives the next observation $o_{i,t+1}$ and the team reward $r_{t+1} = R(s_{t+1})$. For the cooperative multi-agent task, the ultimate goal is to optimize the joint policy $< \pi_1, ..., \pi_n >$ to maximize the $\gamma$ discounted accumulated reward with time horizon $T$: $\mathbb{E}_{\vec{a}_t \sim < \pi_1, ..., \pi_n >} \left[ \sum_{t=1}^{T} \gamma^t r_t \right]$.

**Hierarchical MMDPs.** Considering the hierarchical structure of the coverage problem, we decompose it into two tasks: high-level coordination and low-level execution. The high-level agent (*coordinator*) focuses on coordinating $n$ low-level agents (*executors*) in the long-term to maximize the accumulated team reward $\sum_{t=1}^{T} \gamma^t r_t$. To do it, the *coordinator* $\pi^H(\vec{g}_t|\vec{o}_t)$ distributes goals $\vec{g}_t = (g_{0,t}, g_{1,t}, ..., g_{n,t})$ to the *executors*, based on the joint observation $\vec{o}_t$ ( collected from executors). After receiving the goal $g_{i,t}$ at time step $t$, the executor $i$ locally accomplishes the goal for $k$ steps, i.e., maximizing the cumulative goal-conditioned reward $r_{i,t}^L = R^L(s_t, g_{i,t})$, by continuously taking primitive actions $a_i$ based on the policy $\pi_i^L(a_{i,t}|o_{i,t})$. Since the coordinator interacts with

executors every $k > 1$ steps, the high-level transition could be regarded as a semi-MDP [16]. And the executors still run on a decentralized style as the Dec-POMDP. Differently, the reward function and policy of each are directed by its goal $g_{i,t}$ introduced in the hierarchy. Thus, the semi-MDP and Dec-POMDPs form a two-level hierarchy for multi-agent decision making, referring as a hierarchical Multi-agent MDPs(HMMDPs).

**Attention Modules.** Attention modules [17, 18] have attracted intense interest due to the great capability in a lot of different tasks [19–21]. Furtherly, the self-attention module can handle variably-sized inputs in an order-invariant way. In the paper, we adopt the scaled dot-product attention [17]. Specifically, the matrix of output $\mathbf{H}$ is a weighted sum of the values, which is computed as:

$$\mathbf{H} = Att(\mathbf{Q}, \mathbf{K}, \mathbf{V}) = softmax(\frac{\mathbf{Q}\mathbf{K}^T}{\sqrt{d_k}}) \odot \mathbf{V} \tag{1}$$

where $d_k$ is the dimension of a key; the matrix $\mathbf{K}, \mathbf{Q}, \mathbf{V}$ are the keys, queries, and values, transformed from input matrix $\mathbf{X}$ by parameter matrices $\mathbf{W}_q, \mathbf{W}_k, \mathbf{W}_v$. They are computed as:

$$\mathbf{Q} = tanh(\mathbf{W}_q\mathbf{X}), K = tanh(\mathbf{W}_k\mathbf{X}), \mathbf{V} = tanh(\mathbf{W}_v\mathbf{X}) \tag{2}$$

The context feature $C = \sum_{i=1}^{N} h_i$ summarizes elements in $\mathbf{H}$ in an additive way, where $h_i$ and $N$ are the element and the total number of elements in $\mathbf{H}$.

**Approximate Marginal Contribution.** In the cooperative game, the marginal contribution $\varphi_{C,i}(s)$ of the member $i$ in a coalition $C$ is the incremental value brought by the joining of member $i$. Formally, it is $\varphi_{C,i}(v) = v(C \cup \{i\}) - v(C)$, where $v(\cdot)$ represents the value of a coalition. In a $N$ player setting, Shapley value[22] measures the average of marginal contributions of member $i$ in all possible coalitions, written as $\sum_{C \in N \setminus i} \frac{|C|!(N-|C|-1)!}{N!} \varphi_{C,i}(v)$. $N \setminus i$ denotes the subset of $N$ consisting of all the players except member $i$. Thus, the contributions made by every member can be calculated, once all the sub-coalition contributions $v(C)$ are given. However, it is infeasible to calculate it in practice, as the number of all possible coalitions will be expanded with increasing members $N$, which causes the computational catastrophe. Hence, [11] introduced a method to approximate the marginal contribution by deep neural networks. In this paper, we approximate the marginal contribution of each pair of sensor-target assignments by neural network for learning a coordinator effectively, rather than estimate the marginal contribution of each player.

## 3 Hierarchical Target-oriented Multi-Agent Coordination

Hierarchical Target-oriented Multi-Agent Coordination (HiT-MAC) is a two-level hierarchy, consisting of a coordinator (high-level policy) and $n$ executors (low-level policy), shown as Fig. 1. The coordinator and executors respectively follow the semi-MDP and goal-conditioned Dec-POMDPs in HMMDPs. Periodically, the coordinator aggregates the observations $\vec{o} = (o_1, o_2, \ldots, o_n)$ from the executors and distributes a target-oriented goal $\vec{g} = (g_1, g_2, \ldots, g_n)$ to them. After receiving $g_i$, the executor $i$ will minimize the average angle error to the assigned targets by rotation for $k$ steps based on its policy $\pi_i^L(a_i|o_i, g_i)$. The framework is target-oriented in three-folds: 1) the observation $o_i$ describes the spatial relations among sensor $i$ and all targets $M$ in the environment; 2) the $g_i$ explicitly identifies a subset of targets $M_i \subseteq M$ for the executor $i$ to focus on. 3) the rewards for both levels are highly dependent on the spatial relations among sensors and targets, i.e, the team reward is about the overall coverage rate of targets, the reward for the executor $i$ is about the average angle error between the executor $i$ and its assigned targets.

In the following, we will introduce the key ingredients for HiT-MAC in details.

### 3.1 Coordinator: Assigning Targets to Executors

The coordinator seeks to learn an optimal policy $\pi^{H*}(\vec{g}|\vec{o})$ that can maximize the cumulative team reward by assigning appropriate targets $\{M_i\}_{i \in N}$ for each executor $i \in N$ to track. Note that the coordinator only runs periodically (every $k$ steps) to wait for the low-level execution and save the cost in communication and computation.

**Team reward function** $r_t^H$ for the coordinator is equal to the target coverage rate $\frac{1}{m}\sum_{j=1}^{m} I_{j,t}$ if any target covered (Condition a). $I_{j,t}$ represents the covering state of target $j$ at time step $t$, where 1

is being covered and 0 is not. Notably, if none of targets is covered (Condition b), we will give an additional penalty in the reward. The overall team reward is shown as following:

$$r_t^H = R(s_t) = \begin{cases} \frac{1}{m}\sum_{j=1}^{m} I_{j,t} & (a) \\ -0.1 & (b) \end{cases} \qquad (3)$$

The coordinator is implemented by building a deep neural network, which is composed of three parts: state encoder, actor, critic. There are mainly two challenges to build the coordinator. First, the shapes of the joint observation $\vec{o}$ and goal $\vec{g}$ depend on the number of sensors and targets in the environment. Second, it is inefficient to explore target-assignments only with a team reward directly, especially when the goal space is expanded with the increasing number of sensors and targets. Thus, the network should be capable of 1) handling the variably-sized input and output; 2) finding an effective approach for the critic to estimate values.

**State encoder** adopts the self-attention module to encode the joint observation $\vec{o} \in [o_{i,j}]_{n \times m}$ to an order-invariant representation $\mathbf{H} \in \mathbb{R}_{n \times m \times d_{att}}$. Note that $o_{i,j}$ is a $d_{in}$ dimensional vector, indicating the spatial relation between sensor $i$ and target $j$. In our setting, $o_{i,j} = (i, j, \rho_{ij}, \alpha_{ij})$, where $\rho_{ij}$ and $\alpha_{ij}$ are the relative distance and angle respectively. Please refer to Sec. 4.1 for more details. To feed $\vec{o}$ into the attention module, we flatten it from $\mathbb{R}_{n \times m \times d_{in}}$ to $\mathbb{R}_{nm \times d_{in}}$, then encode it as $\mathbf{H} = Att(\mathbf{Q}, \mathbf{K}, \mathbf{V})$, where $\mathbf{Q}, \mathbf{K}, \mathbf{V}$ are derived from the flatten observation according to Eq. 2.

**Actor** adaptively outputs the goal map $\vec{g} \in \mathbb{N}_{n \times m}$ according to $\mathbf{H} \in \mathbb{R}_{nm \times d_{att}}$. Firstly, we reshape $\mathbf{H}$ as $[n, m, d_{att}]$ again, and compute the probability $p_{ij}$ of each assignment by one fully connected layer, $p_{ij} = f_a(\mathbf{H}_{i,j})$. Then, we sample the assignment $g_{i,j}$ by probability. $g_{i,j}$ is a binary value, indicating if let sensor $i$ to track target $j$. At the end, the actor outputs the goal map $\vec{g}$ for executors, where $g_i = (g_{i,1}, g_{i,2}, ..., g_{i,j})$ denotes the targets assignment for the sensor $i$.

**Critic** learns a value function, which is then used to update the actor's policy parameters in a direction of performance improvement. Rather than directly estimating the global value by a neural network, we introduce an approximate marginal contribution (AMC) approach for learning the critic more efficiently. Similar to most multi-agent cooperation problems, we deduce the individual contribution of each member to the team's success, referred as credit assignment. Differently, we regard each sensor-target pair of the assignments, instead of the agent, as a member of the team. It is because that the coordinator undertakes all the sensor-target assignments, which will directly affect global rewards (if the executors are perfect). Identifying the contribution of each sensor-target assignment to the team reward will be beneficial for a reasonable and effective coordination policy, and such a policy leads to better cooperation among the executors.

Inspired by [11], we approximate the marginal contribution of each assignment (assigning target $j$ to agent $i$) by neural network $\phi$. The input is $\mathbf{H} \in \mathbb{R}_{nm \times d_{att}}$ from the state encoder. The length of $\mathbf{H}$ is $l = nm$, then it can be regarded as a $l$-member cooperation. So, the marginal contribution is approximated as $\varphi_e = \phi([\eta_e, z_e])$. Here $[\cdot, \cdot]$ denotes concatenation, $\eta_e$ is the embedded feature of the sub-coalition $C_e = \{1, ..., e-1\}$ for the member $e$. For example, if the grand coalition is $[z_1, z_2, z_3, z_4]$, then the $\eta_3$ is the context feature of $[z_1, z_2]$, which is used for computing the marginal contribution of the member 3. So, the credit assignment is conducted among all the pairwise sensor-target assignment in the coordinator as Alg.1.

---

**Algorithm 1:** Estimate team value with AMC

**Input:** the state representation $\mathbf{H} \in \mathbb{R}_{nm \times d_{att}}$
**Output:** estimated global team value $v^H$

1 Initialize the sub-coalition feature $\eta_1 = \mathbf{0}$
2 Given an attention module $Att'(\cdot)$ and a value network $\phi(\cdot)$
3 l = n*m
4 **for** *e=1 to l* **do**
5      Compute the marginal contribution $\varphi_e = \phi([\eta_e, h_e])$, where $h_e$ is $e$-th element in $\mathbf{H}$
6      Compute element-wise features of the sub-coalition $\mathbf{H}' = Att'(\mathbf{H}[1:e])$
7      Compute the embedded feature $\eta_{e+1} = \sum_{i=1}^{e} h_i'$, where $h_i'$ is the $i$-th element in $\mathbf{H}'$
8 **end**
9 The team value $v^H = \sum_{e=1}^{l} \varphi_e$

---

Our AMC is conducted on value $v^H$, which is different from SQDDPG [11]. It is because that AMC is conducted on value $Q$ in SQDDPG [11], which would introduce an extra assumption, i.e. the actions taken in $C$ should be the same as the ones in the coalition $C \cup \{i\}$, detailed in Appendix 6.1. Our global value estimation is also different from the existing methods, like [5, 23], because ours refers the sub-coalition contribution to make a more confident estimation of the contribution from each member. Theoretically, the permutation of the coalition formation order should be sampled like the computation of Shapley Value[22]. However, we observe that the permutation of hidden states is useless in our case. And the promotion caused by permutation is also not obvious enough shown in [11]. So, we fix the order in the implementation, i.e. from 1 to $l$.

### 3.2  Executor: Tracking Assigned Targets

After receiving the goal $g_i$ from the coordinator, the executor $\pi_i^L(a_i|o_i, g_i)$ completes the goal-conditional task independently. In particular, the goal of executor $i$ is tracking a set of assigned targets $M_i$, i.e, minimize the average angle error to them.

For training, we further introduce a **goal-conditioned reward** $r_{i,t}^L(s_t, g_t)$ to evaluate the executor. We score the tracking quality of the assigned targets based on the average relative angle, referring to Eq. 4. We consider two conditions, which are (a) the target $j$ is in the coverage range of the sensor $i$, i.e. $\rho_{ij,t} < \rho_{max} \& |\alpha_{ij,t}| < \alpha_{max}$; (b) target is outside of the range. Here $\alpha_{max}$ is the maximum viewing angle of the sensor, $\alpha_{ij,t}$ is the relative angle from the front of the sensor to the target $j$.

$$r_{i,t}^L = \frac{1}{m_i} \sum_{j \in M_i} r_{i,j,t} - \beta cost_{i,t}, r_{i,j,t} = \begin{cases} 1 - \frac{|\alpha_{ij,t}|}{\alpha_{max}} & (a) \\ -1 & (b) \end{cases}, cost_{i,t} = \frac{|\delta_{i,t} - \delta_{i,t-1}|}{z_\delta} \tag{4}$$

where $M_i$ is a set of targets selected for the sensor $i$ according to $g_i$; $cost_{i,t}$ is the power consumption, measured by the normalized moved angle $\frac{|\delta_{i,t} - \delta_{i,t-1}|}{z_\delta}$; $\delta_i$ represents the absolute orientation of sensor $i$, the cost weight $\beta$ is 0.01 and $z_\delta$ is the rotation angle, that is $5°$ in our setting.

**Goal-conditioned filter** is introduced to directly remove the unrelated relations based on the assigned goal firstly. With such a clean input, the executor will not be distracted by the irrelevant targets anymore. For example, if $g_i$ is $[1, 0, 1]$ and $o_i$ is $[o_{i,1}, o_{i,2}, o_{i,3}]$, then $\hat{o}_i = filter(o_i, g_i) = [o_{i,1}, o_{i,3}]$. In other words, the target-oriented goal can be seen as a kind of hard attention map, forcing the executor only to pay attention to the selected targets.

The network architectures of state encoder, actor and critic are detailed in Appendix 5. The action $a_{i,t}$ is the primitive action and the value $v_{i,t}^L$ estimates the coverage quality of the assigned targets of the sensor $i$. All the executors share the same network parameters.

### 3.3  Training Strategy

Similar to most hierarchical RL methods, we adopt the two-step training strategy for stability. It is because that a stochastic executor will lead to a poor team reward, which will bring additional difficulty for learning coordination. At the same time, the coordinator would generate a lot of meaningless goals, e.g. selecting two targets that are far away from each other, which will make the executor confused and waste time on exploration. Instead, the two-step training can prevent the learning of coordinator/executor from the disturbance of the other.

As for the training of the executor, a goal generation strategy is introduced for training the executor without coordinator. Every $k = 10$ time steps, we generate the goal, according to the distance between targets and sensors. To be specific, the targets, whose distances to sensor $i$ are less than the maximum coverage distance ($\rho_{ij,t} < \rho_{max}$), will be selected as the goal $g_{i,t}$ for sensor $i$. Although such strategy mixes some improper targets in the $g_{i,t}$, this will induce a more robust tracking policy for the executor. With the generated goal, We score the coverage quality of the assigned targets for each executor as the individual reward, refer to Eq. 4. Then, the policy can be easily optimized by the off-the-shelf RL method, e.g. A3C [2].

After that, we train the coordinator cooperating with well-performed executors. While learning, the coordinator updates the observation $\vec{o}$ and goal $\vec{g}$ every $k$ steps. During the interval, executor $i$ will take primitive actions $a_i$ step-by-step directed by $g_i$. We fix $k = 10$ in the experiments, while learning an adaptive termination (dynamic $k$) is our future work. The policy is also optimized by A3C [2]. We notice that directly applying the executor learned in the previous step will lead to the large decrease

of the frame rate (only 25 FPS), which causes the training of the coordinator time-consuming. As an alternative, we build a scripted executor to perform low-level tasks to speed up the training process. The scripted executor could access the internal state for designing a simple yet effective programmed strategy, detailed in Appendix 2. Then, the frame rate for the coordinator increases to 75 FPS. Notably, while testing, we use the learned executor to replace the scripted executor, since the internal state is unavailable in real-world scenarios.

## 4 Experiments

First, we build a numerical simulator to imitate the target coverage problem in real-world DSNs. Second, we evaluate HiT-MAC in the simulator, comparing with three state-of-the-art MARL approaches (MADDPG[4], SQDDPG[11] and COMA[6]) and one heuristic centralized method (Integer Linear Programming, ILP) for this problem. We also conduct an ablation study to validate the contribution of the attention module and AMC in the coordinator. Furthermore, we evaluate the generalization of our method in environments with different numbers of targets and sensors. The code is available at `https://github.com/XuJing1022/HiT-MAC` and the implementation details are in Appendix 5.

### 4.1 Environments

To imitate the real-world environment, we build a numerical environment to simulate the target coverage problem in DSNs. At the beginning of each episode, the $n$ sensors are randomly deployed. Meanwhile, the $m$ targets spawn in arbitrary places and walk with random velocity and trajectories.

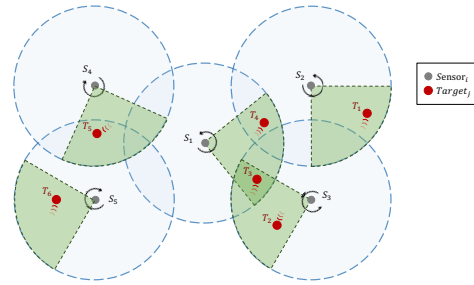

Figure 2: An example of the 2D environment.

**Observation Space.** In every time step, the observation $o_i$ is packed by the sensor-target relations, i.e. $o_i = (o_{i,1}, o_{i,2}, ..., o_{i,m})$. $o_{i,j} = (i, j, \rho_{ij}, \alpha_{ij})$ describes the spatial relation between sensor $i$ and target $j$ in a polar coordinate system (the sensor $i$ is at the origin). To be specific, $i$, $j$ are the ID of the sensor and target separately; $\rho_{ij}$ and $\alpha_{ij}$ are the absolute distance and relative angle from $i$ to $j$. For the coordinator in HiT-MAC, it takes $\vec{o} = (o_1, ..., o_n)$ as the joint observation.

**Action Space.** The primitive action space is discrete with three actions $TurnRight$, $TurnLeft$ and $Stay$. Quantitatively, the $TurnRight/TrunLeft$ will incrementally adjust the sensor's absolute orientation $\delta_i$ in 5 degree, i.e., $Right: \delta_{i,t+1} + = 5$, $Left: \delta_{i,t+1} - = 5$. For the coordinator in HiT-MAC, the goal map $\vec{g}$ is a $n \times m$ binary matrix, where $g_{i,j}$ represents whether the target $j$ is selected for the sensor $i$ (0: No, 1: Yes). Each row corresponds to the assignment for each sensor one by one.

### 4.2 Evaluation Metric

We evaluate the performance of different methods on two metrics: coverage rate and average gain. **Coverage rate (CR)** is the primary metric, measuring the percentage of the covered targets among all the targets, shown in Eq. 3; **Average gain (AG)** is an auxiliary metric to measure the efficiency in power consumption. It counts how much CR gains each rotation brings, i.e. $CR/\widehat{cost}$, where $\widehat{cost} = \frac{1}{Tn} \sum_{t=1}^{T} \sum_{i=1}^{n} cost_{i,t}$, and $cost_{i,t}$ is previously introduced in Eq. 4.

For good performance, we expect both metrics to be high. In practice, we consider the CR in primary. Only when methods achieve comparable CR, AG is meaningful. For mitigating the bias caused by randomness of training and evaluation, we count the results and draw conclusions after running training for 3 times and evaluation for 20 episodes.

### 4.3 Baselines

We employ MADDPG[4], SQDDPG[11] and COMA[6], three state-of-the-art MARL methods as baselines. They all are trained with a centralized critic and executed in a decentralized manner. But their critics are built in different ways for credit assignment, e.g., SQDDPG[11] aims at estimating the shapley Q-value for each agent. As for the target coverage problem in DSNs, one heuristic method is to formulate the problem as an integer linear programming (ILP) problem and globally optimize it at each step. See Appendix 3&4 for more details.

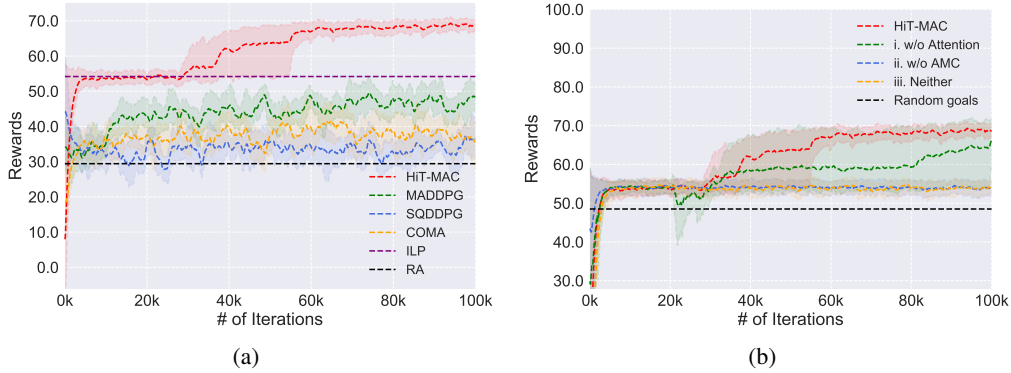

|     |     |
| --- | --- |
| (a) | (b) |

Figure 3: The learning curve of all learning-based methods. They all are trained in environment with 4 sensors and 5 targets. (a) comparing ours with baselines; (b) comparing ours with its ablations.

## 4.4 Results

**Compare with Baselines.** As Fig. 3(a) shows, our method achieves the highest global reward in the setting with 4 sensors and 5 targets. We also draw the mean performance of ILP and random agents in Fig. 3(a) as reference. We can see that state-of-the-art MARL methods work poorly in this setting. None of them could exceed the ILP. Typically, the

Table 1: Comparative results of different methods (n=4&m=5).

| Methods | Coverage rate( % ) ↑ | Average gain( % ) ↑ |
| --- | --- | --- |
| MADDPG | 45.56±9.45 | 1.38 |
| SQDDPG | 36.67±9.04 | 2.73 |
| COMA | 35.37±8.41 | 2.49 |
| ILP | 54.18±12.32 | **3.87** |
| HiT-MAC(Ours) | **72.17±5.58** | 1.46 |

improvement of SQDDPG is marginal to agents with random actions. This suggests that it is difficult to directly estimate the marginal contribution of each agent in this problem. Instead, HiT-MAC surpasses all the baselines after $35k$ of iterations, and reaches a stable performance of $\sim 70$ at the end. As for the quantitative results during evaluation in Tab. 1, HiT-MAC consists of the trained coordinator and trained executors and significantly outperforms the baselines in CR. ILP gets the highest AG, as it globally optimizes the joint policy step-by-step. COMA and SQDDPG also get higher AG than ours, but in fact, they only learn to take no-operation to wait for the targets run into its coverage range. As a result, their CRs are lower than others.

**Ablation Study.** We consider ablations of our method that help us understand the impact of attention framework and global value estimation by AMC shown in the Fig. 3(b). We compare our method with (i) the one without attention encoder; (ii) the one without AMC; (iii) the one without attention encoder nor AMC in $n = 4\&m = 5$. For (i) and (iii), we use bi-directional Gated Recurrent Unit (BiGRU [24, 25]) to replace the attention module. As for the critic input, we use context feature $C_t$ for (ii) and the hidden state of BiGRU for (iii). From the learning curve, we can see that the ablations that without AMC (ii, iii) stuck in a locally optimal policy. Their rewards are close to the random policy, which randomly samples the targets as the goal for executors. Instead, the performance of (i) and ours could be further improved after $35k$ of iterations. This evidence demonstrates that the introduced AMC method is capable of effectively guiding the coordinator to learn a high-quality target assignment. Compared with (i), our method with attention-based encoder converges faster and more stably. And the variance of the training curve in (i) is larger than ours, though the one without attention-based encoder can also converge to a high score sometimes. So, we think that the attention-based encoder is more suitable for the coordinator, rather than RNN. It is because that attention mechanism can aggregate the features without any assumption about the sequence order, rather than following a specific order to encode the data.

**Generalization.** We analyze the generalization of our method to the different number of sensors $n$ and targets $m$. While

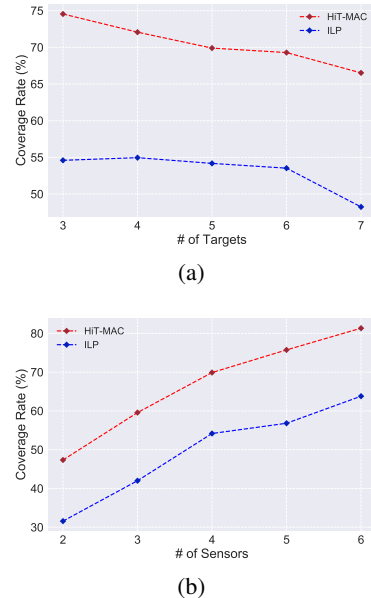

(a)

(b)

Figure 4: Analyzing the generalization of HiT-MAC to the different number of sensors $n$ and targets $m$. (a) $n = 4$, $m$ is from 3 to 7; (b) $m = 5$, $n$ is from 2 to 6.

testing, we adjust the number of targets and sensors in the environment, respectively, and report the mean coverage rate under each setting for better analysis. For example, in Fig. 4(a), we can see the trend of performance with the change of the target number from 3 to 7 in the 4 sensors case. In the same way, we also demonstrate the trend of performance in the cases of varying sensor numbers ($n = 2, 3, 4, 5, 6 \& m = 5$), shown as Fig. 4(b). Note that our model is only trained in a fixed-number environment ($n = 4 \& m = 5$). We report the rewards of ILP as reference, as its performance does not depend on the training environment and has stable generalization in different settings. Since the score of ILP is already lower than ours, we compare the changing of the score to ensure as much fairness as possible. In Fig. 4(a), the performance of ours increases stably with the decrease of $m$ from 5 to 3, while the reward of ILP increases lightly. With the increase of $m$ from 5 to 7, ours decreases slower than ILP. In Fig. 4(b), our score increases more stably than ILP when $n$ increases from 4 to 6. Those can be concluded that HiT-MAC is scalable and of a good generalization in environments with different numbers of sensors and targets.

## 5   Related Work

**Coverage Problem** is a crucial issue of directional sensor networks [26]. The available studies about coverage problem can be categorized into four main types [27]: target-based coverage, area-based coverage, sensor deployment, and minimizing energy consumption. And a set of heuristic algorithm [28–30] has been proposed to find a nearly-optimal solution under a specific setting, as most of the problems are proved NP-hard. Recently, with the advances of machine learning, Mohamadi, et al. adopt learning-based methods [31, 32] for maximizing network lifetime in wireless sensor networks. However, all the algorithm are designed for a specific setting/goals. In this work, we focus on finding a non-trivial learning approach for the target-based coverage problem. We formulate the coverage problem as a multi-agent cooperate game, and employ the modern multi-agent reinforcement learning to solve the game.

**Cooperative Mutli-Agent Reinforcement Learning(MARL)** addresses the sequential decision-making problem of multiple autonomous agents that operate in a common environment, each of which aims to collaboratively optimize a common long-term return [33]. With the recent development of deep neural network for function approximation, many prominent multi-agent sequential decision-making problems are addressed by MARL, e.g. playing real-time strategy games [7], traffic light control [8], swarm system[9], common-pool resolurce appropriation [34], sequential social dilemmas [35], etc. In Cooperative MARL, it is notoriously difficult to give each agent an accurate contribution under a shared reward. This phenomenon limits the further application of MARL in more difficult problems, referred as credit assignment. This motivates the study on the local reward approach, which aims at decomposing the global reward to agents according to their contributions. [10, 6] modeled the contributions inspired by the reward difference. Based on shapley value [36], shapley Q-value [11] is proposed to approximate a local reward, which considers all possible orders of agents to form a grand coalition. In this paper, we learn the critic in coordinator by approximating the marginal contribution of each sensor-target assignment for effective learning the coordination policy.

## 6   Conclusion and Discussion

In this work, we study the target coverage problem, which is the main challenge problem in the DSNs. We propose an effective Hierarchical Target-orient Multi-agent Coordination framework (HiT-MAC) to further enhance the coverage performance. In HiT-MAC, we decompose the coverage problem into two subtasks: assigning targets to sensors and tracking assigned targets. To implement it, we further introduce a bunch of practical methods, such as AMC for the critic, attention mechanism for state encoder. Empirical results demonstrated that our method can deal with different scenes and outperform the state-of-the-art MARL methods.

Although significant improvements have been achieved by our methods, there is still a set of drawbacks waiting for addressed. 1) We need to find a solution to deploy the framework in large scale DSNs ($n > 100$), e.g. multi-level hierarchy. 2) For a practical application, it is necessary to additional consider a more real-world setting, including placing obstacles, using visual observation, and limited communication bandwidth. 3) For the executor, it is required to learn an adaptive termination, rather than the fixed k-step execution. Furthermore, it is also an interesting future direction to apply our method to other target-oriented multi-agent problems, where agents are focused on optimizing some relations to a group of targets, e.g. collaborative object searching[4], active object tracking [37].

## Broader Impact

The target coverage problem is common in Directional Sensor Networks. This problem widely exists in a lot of real-world applications. For example, those who control the cameras to capture the sports match videos may benefit from our work, because our framework provides an automatic control solution to free them from heavy and redundant labor. Surveillance camera networks may also benefit from this research. But there is also the risk of being misused in the military field, e.g., using directional radar to monitor missiles or aircraft. The framework may also inspire the RL community, for solving the target-oriented tasks, e.g. collaborative navigation, Predator-prey. If our method fails, the targets would be all out of views of sensors. So, maybe a rule-based alternate plan is needed for unexpected conditions. We reset the training environment randomly to leverage biases in the data for better generalization.

## Acknowledgments and Disclosure of Funding

We thank Haifeng Zhang, Wenhan Huang, and Prof. Xiaotie Deng for their helpful discussion in our early work. This work was supported by MOST-2018AAA0102004, NSFC-61625201, the NSFC/DFG Collaborative Research Centre SFB/TRR169 "Crossmodal Learning" II, Qualcomm University Research Grant, Tencent AI Lab RhinoBird Focused Research Program (JR201913).

## Footnotes

* indicates equal contribution

[1]https://playsight.com/automatic-production/

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
