[Supplementary Material]

# Appendix

## 1 Goal generation for executor training

The pseudo goal generation is introduced for training the executor without coordinator. Every 10 time steps, we generate the goal, according to the distance between targets and sensors. To be specific, the targets, whose distances to sensor $i$ are less than the maximum coverage distance ($\rho_{ij,t} < \rho_{max}$), will be selected as the goal $\vec{g_{i,t}}$ for sensor $i$. Although the generated goals are not aligned to the policy of the ideal coordination, some improper targets mixed in the $\vec{g_{i,t}}$ can lead the executor to learn a more robust policy to track the assigned stargets. Then, such a robust executor may adapt to the coordinator trained in the next stage better.

## 2 Programmed strategy for the executors

we build a scripted policy to perform low-level tasks while training the coordinator, since directly applying the learned executor will cause the frame rate to drop severely (only 25 FPS). The scripted policy is allowed to access the grounded state, e.g. the absolute position $(x_j, y_j)$ of each targets and the pose of the sensor. Intuitively, one feasible solution to track a set of targets is to make the sensor pointing to the cluster center of the targets. Thus, we calculate the center $(x_{mean}^M, y_{mean}^M)$ of the assigned targets $M_i$ for sensor $i$, while the script policy is to take primitive actions to minimize the relative angle to the center. Given the pose of sensor $i$ is $(x_i, y_i, \alpha_i)$ where $(x_i, y_i)$ is the position and $\alpha_i$ is the orientation, the relative angle error $\beta_i$ is calculated as

$$arctan(\frac{y_{mean}^M - y_i}{x_{mean}^M - x_i}) * \frac{180}{\pi} - \alpha_i \qquad (1)$$

And, the taken action $a_{i,t} = clip(\beta_i // z_\delta, -1, 1)$. Here $z_\delta = 5$, because the action is deterministic and the rotation unit is 5 degree. With this trick, the frame rate for training of the coordinator increases to 75 FPS. Note that it is not the optimal policy for the executor, it will fail when two targets are far.

## 3 Integer Linear Programming

As one of the baselines, we formulate the problem as an Integer Linear Programming (ILP) problem and solve it by CBC[1] (Coin-or branch and cut) optimizer. The inputs are position, rotation, sensing range of sensors and position of targets, while the outputs are the primitive actions taken by every sensor, i.e. $TurnRight$, $TurnLeft$ or $Stay$.

The notations used here are defined as follows.

- $N$: the number of the sensors
- $P$: the number of optional directions for a sensor
- $S$: the set of sensors
- $s_i$: the i-th sensor

Figure 1: Optional direction partition

- $s_{i,j}$: the j-th direction of the i-th sensor
- $R_{i,j}$: the monitor region of $s_{i,j}$
- $M$: the number of the targets
- $T$: the set of targets
- $t_k$: the k-th target

Consider a randomly deployed directed sensor network, there are $N$ directed sensors $S$ to monitor the targets $T$, and each sensor has $P$ optional directions. First of all, we set P as 4 shown in Fig1, because the visual angle is $90°$. $S_{i,1}$ is always the current direction of the sensor $i$. The other optional directions are clockwise numbered as $S_{i,j}$. And the primitive actions depends on the selected directions solved from ILP.

The variables in the ILP are described as follows: the binary variable $y_k$ is 1 if and only if the target $t_k$ is covered by an arbitrarily sensor, otherwise it is 0; the binary variable $x_{i,j}$ takes the value 1 if and only if the i-th sensor is in the j-th direction, otherwise it takes 0.

For each directed sensor, define a matrix whose element $a^i_{j,k}$ indicates whether the target $t_k$ is in the $s_{i,j}$ sensing area:

$$a^i_{j,k} = \begin{cases} 1, & t_k \in R_{i,j} \\ 0, & otherwise \end{cases} \quad \forall j = 1, \cdots, P; \forall k = 1, \cdots, M \tag{2}$$

Define a non-negative integer $\delta_k$ as the number of sensors that cover the target $t_k$:

$$\delta_k = \sum_{i=1}^{N} \sum_{j=1}^{P} a^i_{j,k} x_{i,j}, \forall k = 1, \cdots, M \tag{3}$$

So, the target coverage problem can be formulated as Integer Linear Problem as follows:

$$\max \ z = \sum_{k=1}^{M} y_k$$

$$s.t. \quad \begin{cases} \delta_k/N \le y_k \le \delta_k, & \forall k = 1, \cdots, M \\ \sum_{j=1}^{P} x_{i,j} \le 1, & \forall i = 1, \cdots, N \\ y_k = 0 \, or \, 1, & \forall k = 1, \cdots, M \\ x_{i,j} = 0 \, or \, 1, & \forall i = 1, \cdots, N; \forall j = 1, \cdots P \end{cases}$$

The objective is to maximize the number of covered targets. The first constraint denotes whether $t_k$ is covered. The second one denotes that a sensor can only work in one of the optional directions at the same time.

After formulation, we can solve the target coverage problem as an ILP problem with CBC optimizer. Then, the primitive actions for all the sensors can be derived from the results of ILP shown as Tab. 1.

Table 1: Derivation rules

| $x_{i,j}$ | primitive action |
| --- | --- |
| $x_{i,1}$=1 | $Stay$ |
| $x_{i,2}$=1 | $TurnRight$ |
| $x_{i,3}$=1 | $TurnRight$ or $TurnLeft$ |
| $x_{i,4}$=1 | $TurnLeft$ |

In the section 4.3, we can see that the performance of ILP is poor. The reasons are three folds.

- 1) Integer programming is better for static allocation, while targets is preferably within the field of view radius. But, the targets are mobile in our settings. So, when targets are outside the field of view, it needs prediction based on sequential history observations.

- 2) The objective function is to maximize the coverage rate, therefore the rotation can only be taken when some target leaves the current field of view, which will lead to lower score. Actually, sensors should be rotated in advance if possible in order to avoid targets loss.

- 3) There may be some solutions for the above two problems. For example, handcraft the objective function with intuition, e.g. additionally considering the average relative angle error to the targets. But manually design the constraints will be trivial and difficult.

## 4  MARL baselines

We implement the MARL baselines by employing the codes from `https://github.com/hsvgbkhgbv/SQDDPG`. To be specific, the policy and critic network both are two layers MLP for MADDPG and COMA, as the same as the policy network and hyper network for mixing Q value in the Q-mix. The common hyper-parameters are detailed in Tab. 2.

Table 2: Hyper-parameters for baselines

| Hyper-parameters | # | Description |
|---|---|---|
| hidden units | 128 | the # of hidden units for all layers |
| training episodes | 50k | maximum training episodes |
| episode length | 100 | maximum time steps per episode |
| discount factor | 0.9 | discount factor for rewards, i.e. gamma |
| entropy weight | 0.001 | parameter for entropy regularization |
| learning rate | 5e-4 | learning rate for all networks |
| target update frequency | 100 | target network updates every # steps |
| target update rate | 0.1 | target network update rate |
| replay buffer | 1e4 | the size of replay buffer |
| batch size | 64 | the # of transitions for each update |

## 5  Networks and Hyper-parameters for HiC-MAC

As for coordinator, the encoder consists of 2 fully connected(FC) layers and an attention module, the actor consists of one FC layer and the critic based AMC consists of an attention module and one FC layer. As for executor, the encoder is an attention module, the actor and the critic both consist of one FC layer simply. The training framework is like A3C, and the hyper-parameters for our method are detailed in Tab. 3.

Table 3: Hyper-parameters for both coordinator and executor.

| Hyper-parameters | # | Description |
|---|---|---|
| hidden units | 128 | the # of hidden units for all layers |
| training episodes | 50k | maximum training episodes |
| episode length | 100 | maximum time steps per episode |
| discount factor | 0.9 | discount factor for rewards |
| entropy weight | 0.01 | parameter for entropy regularization |
| learning rate | 5e-4 | learning rate for all networks |
| workers | 6 | the # of workers in the A3C framework |
| update frequency | 20 | the master network updates every # steps in A3C |

# 6 Discussion about AMC

## 6.1 Compare with Shapley Q value in SQDDPG

The marginal contribution of each coalition in SQDDPG is defined as $\Phi_i(\mathcal{C}) = Q^{\pi_{\mathcal{C} \cup \{i\}}}\left(s, \mathbf{a}_{\mathcal{C} \cup \{i\}}\right) - Q^{\pi_{\mathcal{C}}}\left(s, \mathbf{a}_{\mathcal{C}}\right)$. And they model a function to approximate the marginal contribution directly such that $\hat{\Phi}_i\left(s, \mathbf{a}_{\mathcal{C} \cup \{i\}}\right) : \mathcal{S} \times \mathcal{A}_{\mathcal{C} \cup \{i\}} \mapsto \mathbb{R}$, where $\mathcal{S}$ is the state space; $\mathcal{C}$ is the ordered coalition that agent $i$ would like to join in; $\mathcal{A}_{\mathcal{C} \cup \{i\}} = (\mathcal{A}_j)_{j \in \mathcal{C} \cup \{i\}}$ and the actions are ordered.

In SQDDPG, AMC is conducted on $Q(s, a)$ value, which would introduce an extra assumption, i.e. the actions taken in $C$ should be the same as the ones in the coalition $C \cup \{i\}$. In fact, the actions of every agent in different coalitions are not necessarily the same. Here is an example. Given three agents cooperating to complete a task, their optimal joint action is $\mathbf{a} = (a_0, a_1, a_2)$. If the order of a coalition is (0,2), then $\mathbf{a}_{\mathcal{C} \cup \{1\}} = (a_0, a_2, a_1)$. However, if there are just $agent_0$ and $agent_2$ that cooperate in the environment, the optimal actions $(\hat{a}_0, \hat{a}_2)$ for coalition (0,2) may be different from $(a_0, a_2)$. Actually, the marginal contribution should be $\Phi_i(\mathcal{C}) = Q^{\pi_{\mathcal{C} \cup \{i\}}}\left(s, (a_0, a_1, a_2)\right) - Q^{\pi_{\mathcal{C}}}\left(s, (\hat{a}_0, \hat{a}_2)\right)$. So, it is infeasible to construct different sub-coalitions by just reordering actions. If someone wants to apply Shapley Value or marginal contribution to Q value, the optimal actions for sub-coalitions need to be available with a certain method. Instead, our AMC is conducted on state value $v^H$ that is not directly related with specific actions, which is different from SQDDPG.

## 6.2 The effectiveness of AMC in different goal space

Figure 2: Figures prove that the effectiveness of AMC is reflected as the complexity of the environment rises, i.e. more sensors and targets. The two figures from left to right are the experiments when $n = 2\&m = 3$ and $n = 4\&m = 5$, where n is the number of sensors and m is the number of the targets.

The results from the paper show the contribution of AMC. However, whether is AMC critical all the time? We study the situation that the contribution of Approximate Marginal Contribution(AMC) are more apparent. The studies are conducted in $n = 2\&m = 3$ and $n = 4\&m = 5$. In the simple setting 2(a), i.e. 2 sensors cover 3 targets, we find that the one without AMC can already obtain a good performance. And the convergences are similar in this setting. But when the setting becomes complex, our advantages appear. When there are 4 sensors and 5 targets in the environment in Figure 2(b), the one without AMC can even not converge to a local optimal solution sometimes, since its policy entropy stay high and can not decrease. Sometimes, the one without AMC can obtain a good score, while the convergence is slower than ours. The variance between several training sessions is large. But, our method outperforms more stably. So, we conclude that global value estimation by AMC is effective and necessary when the cooperative setting becomes complicated.

## Footnotes

[1]https://github.com/coin-or/Cbc