[Reviews · NeurIPS 2020]

Review 1

Summary and Contributions: The authors propose a reinforcement learning approach to the target coverage problems in directional sensor network. The authors introduce a hierarchical multi-agent RL algorithm. The lower level of the hierarchy is the “executor” which is a policy network trained using standard RL while the higher level is a coordination mechanism relying on attention to identify the contribution of the different agent and assign goals for the lower level layer. The authors provide empirical results showing the advantage of their method against state of the art MARL algorithms as well as optimization techniques specific to the target coverage problem.

Strengths: Although the paper is tailored to the target coverage problem, I believe the coordination mechanism could be applied to the larger class of cooperative multi-agent problems. I believe using attention mechanism in MARL is not novel but the framing of the higher level hierarchy estimating the marginal contribution of each agent is original and interesting. The experiments show clearly the advantage of the approach and the choice of the baselines is sound. The author propose an ablation study to identify the benefit of the different components they introduce.

Weaknesses: The paper is targeted toward the specific problem of target coverage. Even if a lot of components could generalize, it is hard to asses from the experiments if that would be the case since only one domain is considered. Even though it is discussed, it would have been interesting to show quantitative results on the impact of the two step training (executor and then coordinator) vs an end to end method. Could you provide some insights? The authors provide a well crafted reward function for the problem of interest, it would be interesting to study the influence of this design on the overall performance. How much would that affect the results? How does the scalability of this method compare with other algorithm in terms of number of targets and number agents? Would the centralized coordinator scale to > 20 agents? This claim: “the critic based on such global feature cannot work well” needs further justification.

Correctness: The methods are correct and the experiments seem to be carried correctly (repeated trials, various baselines, report relevant metrics), except for the training episodes: In the appendix, the author show that they use less training episodes for the baselines than for their method, why? Is that still a fair assessment?

Clarity: The description of the coordinator could be improved. In particular: “ There has been ID in the pairwise observation to distinguish them from each other. Thus, their order becomes not quite meaningful.” How are the agents and target ID? Why wouldn’t the order matter here? The filtering component is barely explained. What is its role? Why is it needed? And how is it implemented? Formatting: L 212: evaluation Labels on figure 2 are not readable

Relation to Prior Work: The authors cite the relevant body of literature and the relation is pretty clear.

Reproducibility: Yes

Additional Feedback: The filtering part needs more details if one wanted to reimplement the algorithm. The author provide code for both the environment and the algorithm. I have read the authors' response and wish to keep my score.


Review 2

Summary and Contributions: The authors are considering a directional sensor network with a number of randomly moving targets. A number of sensors are deployed in the field in fixed locations and are tasked with maximizing the number of targets they capture. The sensors can only capture information in a certain angle. However, it is assumed that the sensors can rotate (change the direction of capture). The authors are formulating this problem as one of multi-agent reinforcement learning. Inspired from the model of Hierarchical Reinforcement Learning, they propose an approach called Hierarchical Multi-agent Coordination Mechanism. The hierarchy consists of a centralized coordinator and the sensors. One of the challenges the authors address in their model is that the large number of candidate combinations for target selection for the individual sensor nodes, which they address with an attention based representation.

Strengths: Overall, the authors show a good understanding of the multi-agent reinforcement learning model and applied it to a concrete application area.

Weaknesses: - It is not clear that the application area calls for a multi-agent model here. If there is a central observer that can observe everything, and the interests of the nodes are exactly aligned, and you have a real time communication: this is essentially the case of centralized control. - The authors essentially only compare with other MARL approaches - and do not give a serious thought of what other alternatives exist. For instance, for the problem sizes the authors discuss (4 sensors and 5 targets) the optimal direction of the sensors can be found trivially through search. The authors also do not discuss whether their approach outperforms trivial distributed heuristics (for instance: each agent should make sure that it tracks the target closest to it, and within this constraint, try to track other targets as well as possible). - It is not clear that the proposed approach would generalize to any other application.

Correctness: As far as I can tell, the claims, method and evaluation approach are correct.

Clarity: Overall, the paper is well written.

Relation to Prior Work: I am not aware of prior work tackling this particular challenge.

Reproducibility: Yes

Additional Feedback: I have read the authors feedback. I appreciate the authors running the simple heuristics I proposed - hopefully the authors agree with me that the fact that a non-learning heuristic outperforming all the other approaches except the one proposed, raise some questions about the structure of the problem. I don't find convincing the answer about the need for MARL - that the coordinator can only communicate every 10 "steps" - where is 10 coming from? Also, how much is a "step" - is is 1ms, 1s, 1min??? Is this for sports videos or for airplane tracking?


Review 3

Summary and Contributions: 1) This paper proposes a Hierarchical Multi-agent Coordination Mechanism for solving the target coverage problem in DSNs. 2) Attention mechanism and approximating marginal contribution (AMC) is utilized to improve the performance. 3) Experiments in simulators illustrate the effectiveness of the proposed method.

Strengths: -Overall, the paper is well written and clear. -The paper presents a hierarchical multi-agent coordination mechanism for the task of enhancing the target coverage in DSNs. -The design choice seems reasonable and its intuition is well motivated. -Ablation studies show the effectiveness of the proposed components. -Details & codes are provided to improve the reproducibility.

Weaknesses: 1) The task of enhancing the target coverage in Directional Sensor Networks (DSNs) is important and challenging. However, as far as I am concerned, it is not a standard benchmark environment for studying multi-agent reinforcement learning. The proposed method/model design targets at a specific problem, limiting its significance. There already exist some popular environments for multi-agent cooperation. If experiments are conducted on these standard benchmarks, the significance of this work for the machine learning (ML) or reinforcement learning (RL) community can be improved. 2) The design choice is reasonable, however, the technical novelty seems a little bit low. Attention-based encoders have been widely used in literature to solve the problem with the variable length. And modeling the marginal contribution seems straightforward. 3) As shown in Table.1, the average gain of the proposed method is not as good as COMA or ILP. Are there any comments?

Correctness: The description of the method is mostly clear. The design choices of the components are well-validated.

Clarity: The submission is clearly written and well organized.

Relation to Prior Work: The related work section is mostly clear and explain the difference with existing approaches.

Reproducibility: Yes

Additional Feedback: Question1: It seems that the authors will release the codes/environments to the public. Can the authors confirm this? ---------------------------------------------------------------------------------- Update: After reading other reviewers' comments and the author responses, I share the reviewers' concerns about the limited application areas, and experimental settings. Regarding my own concerns, the author responses did not really address my concerns about limited novelty and lack of benchmark experiments. The paper targets at solving a specific problem of multiple target coverage using hierarchical multi-agent algorithm. However, since it is not a standard benchmark, the effectiveness of the algorithm is still concerning. I would like to keep my original score.

[Author Response · NeurIPS 2020]

We thank the reviewers for all of these valuable comments. We provide point-by-point responses below.

**Re: generalize to other applications.** The target coverage problem is a fundamental and challenging problem in
most DSNs, covering various practical applications, such as camera networks for sports game videos capturing and
directional radar networks for aircraft tracking. We formulated it as a Multi-agent Markov Dynamic Process and
proposed a coordinator-executor mechanism with a bunch of practical methods. We are grateful for your reminder of
the generalization potential of our approach. Indeed, it is promising to apply the coordinator-executor hierarchy to other
environments with multi-targets and multi-agents. Upon your suggestions, we tried to apply our mechanism in the 3v3
Cooperative Navigation problem (Lowe et al. '17) and achieved a competitive mean reward (-4.8) against MADDPG
(-5.3) and COMMA (-7.7). It is an interesting future direction to apply our method on other multi-target multi-agent
coordination problems.

**[R1] Q1: training method.** We found that simultaneously training the coordinator and executor works poorly, as they
destabilize each other. Specifically, the stochastic target selection will make the executor inefficient to learn. Meanwhile,
the poor executor will lead to a noisy global reward, destabilizing the learning of the coordinator. Instead, the two-step
training strategy could avoid these problems. The coverage rate of the end-to-end method is $36.57 \pm 6.78$, which is
much worse than the two-step training.

**Q2: reward design.** The reward function design is based on the domain knowledge of the DSNs, counting the coverage
rate and the energy cost. We will further discuss the factors of each component in the next revision.

**Q3: scalability.** We have compared the scalability of our method with ILP in the case of 3-7 targets and 2-6 sensors
(See L288-297 and Fig.4(b)(c)). Notably, our model is only trained in a specific setting (4 sensors, 5 targets). If the
evolutionary population curriculum approach (Long et al. '20) is employed while training, the scalability could be
improved further.

**Q4: the critic based on global feature.** This claim is supported by the poor performance of the case (2) in ablation
study, described around L280. Main reason is that the global feature $C_t$ just aggregates the local features $\theta_t^i$ by sum.
While, the critic based on AMC can assign credit more accurately, which leads to a better policy.

**Q5: training episodes for the baselines.** Sorry for the typo. In fact, all methods are equally trained with 50k episodes.

**Q6: clarity:** 1) In that paragraph, we want to argue that the order of relation observations wouldn't matter, because the
relations are pairwise and unordered, not like text data requiring sequential processing. So, the attention mechanism is
better than RNN for the coordinator to encode an order-invariant representation. Then, IDs are just for distinguishing
relations from each other. 2) The role of the **filter** is to remove the redundant target-agent relations from the observation
$o_{i,t}$ of the executor $i$ according to $g_{i,t}$. For example, if $g_{i,t}$ is $[1, 0, 1]$ and $o_{i,t}$ is $[o_{i,t}^1, o_{i,t}^2, o_{i,t}^3]$, then $f(o_{i,t}, g_{i,t}) =$
$[o_{i,t}^1, o_{i,t}^3]$. In this way, the executor can ignore irrelevant targets and focus on the assigned targets. 3) The typos and
formatting error would be corrected in the next revision. Thanks for pointing them out.

**[R2] Q1: It is not clear that the application area calls for a multi-agent model here.** For a sensor network, it is
necessary to consider the communication cost among sensors. Thus, a fully-centralized controller is not a good solution
to this problem. In our mechanism, the coordinator is only allowed to communicate with executors **periodically (every
10 steps)**, instead of every step. And each distributed executor only needs to focus on its sub-task independently without
any additional communication. Besides, the coordinator could pay more attention to long-term schedule.

**Q2: only compare with other MARL approaches?** In fact, we already compared our method with a non-MARL
baseline, namely ILP (Integer Linear Programming). Please refer to Appendix.2 for the implementation details and
further analysis. Such a fully-centralized optimization method requires a strictly problem-specific formulation and some
fine-crafted constraints, but performs worse than our MARL method. This also shows the necessity of a multi-agent
model for this application (mentioned in **R2Q1**). Upon your suggestion, we also ran the trivial distributed heuristics.
The performance [coverage rate: $60.65 \pm 7.91\%$, average gain:1.63] is worse than ours.

**[R4] Q1: standard benchmarks.** In this paper, we aim at finding a solution to address the target coverage problem in
DSNs and new challenges encountered. We evaluate methods on the customized environments as there is no benchmark
that could satisfy our problem setting. If accepted, we will release our environments to enrich the public benchmarks
and attract researchers to pay more attention to the new challenge, i.e. multi-agents multi-targets assignment.

**Q2: the technical novelty.** 1) We first propose a Hierarchical Multi-agent Coordination Mechanism for solving the
target coverage problem in DSNs. There is no prior work tackling this particular challenge via RL (as mentioned by R2).
2) Then, we introduced a bunch of practical and effective methods for the mechanism. Particularly, "The framing of the
higher-level hierarchy estimating the marginal contribution of each agent is original and interesting." (as recognized by
R1) 3) It is also promising to generalize our approach to other domains (as mentioned by R1).

**Q3: the average gain** This is an auxiliary metric to evaluate the efficiency of the energy consumption. It is only useful
when a competitive coverage rate has been achieved. COMA only learns to take no-operation most of the time, so it
saves the energy but also get the worst performance in the coverage rate. ILP prefers to rotate the sensors only when the
targets are being out of the current direction partition as analyzed in Appendix.2, so its average gain is the highest.

**Q4: release the codes/environments?** If accepted, we will release the environments, codes, and trained models.

[Meta-Review · NeurIPS 2020]

This paper proposes multi-agent hierarchical RL method to the target coverage problems in directional sensor networks. Empirical results are provided to show the advantage of their method against state of the art MARL algorithms as well as optimization techniques specific to the target coverage problem. There are some concerns among the reviewers regarding whether RL is the right tool for the problem, insufficient comparison with non-learning heuristics, and the value of the work to the RL community. I share the first reviewer’s positive sentiment on the application of RL to sensor networks. It is nice to see RL moving from games to real-world applications. I also agree with the first reviewer that the proposed method do have promise in comparison with non-learning heuristics and the same approach could be applied to other MA problems.